COMMUNICATIONS

# Genome-wide translational profiling of amygdala *Crh*-expressing neurons reveals role for CREB in fear extinction learning

Kenneth M. McCullough [1], Chris Chatzinakos[1], Jakob Hartmann [1], Galen Missig[1], Rachael L. Neve[2], Robert J. Fenster[1], William A. Carlezon Jr.[1], Nikolaos P. Daskalakis [1,3 ✉] & Kerry J. Ressler [1,3 ✉]

Fear and extinction learning are adaptive processes caused by molecular changes in specific neural circuits. Neurons expressing the corticotropin-releasing hormone gene (*Crh*) in central amygdala (CeA) are implicated in threat regulation, yet little is known of cell type-specific gene pathways mediating adaptive learning. We translationally profiled the transcriptome of CeA *Crh*-expressing cells (Crh neurons) after fear conditioning or extinction in mice using translating ribosome affinity purification (TRAP) and RNAseq. Differential gene expression and co-expression network analyses identified diverse networks activated or inhibited by fear vs extinction. Upstream regulator analysis demonstrated that extinction associates with reduced *CREB* expression, and viral vector-induced increased CREB expression in Crh neurons increased fear expression and inhibited extinction. These findings suggest that CREB, within CeA Crh neurons, may function as a molecular switch that regulates expression of fear and its extinction. Cell-type specific translational analyses may suggest targets useful for understanding and treating stress-related psychiatric illness.

[1] McLean Hospital, Department of Psychiatry, Harvard Medical School, Belmont, MA 02478, USA. [2] Gene Transfer Core, Massachusetts General Hospital, Boston, MA 02114, USA. [3] These authors contributed equally: Nikolaos P. Daskalakis, Kerry J. Ressler. ✉email: ndaskalakis@mclean.harvard.edu; kressler@mclean.harvard.edu

Fear and fear extinction learning are evolutionarily conserved, homeostatic processes that are critically perturbed in a variety of neuropsychiatric disorders, such as posttraumatic stress disorder (PTSD), generalized anxiety disorders, bipolar disorder, and Alzheimer's disease[1–4]. Fear learning is a complex process involving associative learning of explicit trauma-paired cues and more generalized contextual elements[5]. Understanding the neural circuitries regulating this process, and molecular changes in these circuitries following behavior may reveal translationally relevant molecular pathways relevant to the diagnosis, treatment, and even prevention of human disease[6,7]. As an example, PTSD has been repeatedly associated with a failure to recover from traumatic events, conceptualized as a failure to extinguish learned fear[8–10]. Notably, the largest-to-date genome-wide association study (GWAS) for PTSD recently found genome-wide level associations between variants of *CRHR1*, the gene encoding corticotropin-releasing hormone (CRH) receptor-type 1, genetically regulated brain-specific *CRHR1* expression, and PTSD diagnosis and symptom clusters[11,12], along with its identification in large GWAS of anxiety and habitual alcohol use[13,14].

The amygdala is a primary integrator of aversive physical stimuli and associated cues[15,16]. Specifically, the central amygdala (CeA) plays a well characterized role in regulating the expression of defensive responses via its connections to downstream regions[17]. The CeA comprises three cytoarchitecturally distinct sub-compartments (central capsular (CeC), lateral (CeL), and medial (CeM)). Recent characterizations of CeA circuitry have identified multiple recursive inhibitory loops which hierarchically gate threat responses (reflexive, active, passive, etc.) through their intra-CeA and output projections[18–22]. Within the CeA there are several molecularly distinct neuronal populations which mediate specific elements of fear conditioning (FC) and extinction (EXT) learning processes[23–33]. Specifically, within the CeL, three populations represent the majority of neurons: those marked by *Crh*, *Sst*, or *Prkcd*,[17,22,23,32]. Until recently, it has not been possible to characterize translational regulation of mRNAs selectively within each of these cell populations, this enables identification highly specific processes that could be targeted for diagnostics or therapeutics.

Hypothalamic CRH is a critical initiating signal of the stress response that is converted to a whole-body stress response via the hypothalamus–pituitary–adrenal (HPA) axis[34–38]. Within the CeL, Crh neurons are critical mediators of the aversive stimulus response, as well as of associative learning[17,39–41]. Recently, functional dissection of Crh neuronal activity during FC and EXT demonstrated a critical role for Crh neurons in the acquisition of associative fear, as well as the acquisition of EXT[40,42–44]. Crh neuronal activity is necessary for the acquisition of weak threats, whereas these neurons appear to be silenced during extinction of learned fear memories[40,42]. In addition, a mutually inhibitory circuit between Crh and Sst neurons gate the generation of active and passive fear responses[17].

FC and EXT precipitate changes in the expression of genes within distinct cellular populations responsible for long-lasting memory. Given the critical role the CeA Crh neuron population plays in fear learning and memory, characterizing the molecular changes specifically within this population during FC and EXT learning may provide valuable insight into potential therapeutic interventions for humans with fear-related disorders[45]. Here, using translating ribosome affinity purification (TRAP) and RNA sequencing (TRAP-seq)[46,47], we examine changes in polysome-associated RNAs of Crh neurons within the CeL, following tone alone (TA), FC, or EXT in both male and female mice[47,48]. We perform gene co-expression network analysis[49], followed by gene set enrichment analysis (GSEA)[50] and upstream regulator

analysis (URA)[51] of gene and gene network changes to identify pathways regulated by EXT in the CeL Crh population, identifying CREB (cAMP response element binding protein) as an upstream regulator of gene expression during EXT in Crh neurons[52–56]. By validating genes and networks, together with the use of cell type-specific viral-mediated gene transfer for causal analyses, we discovered that CREB levels in Crh neurons regulate fear expression and the efficacy of EXT learning. Our findings demonstrate distinct patterns of translational change following EXT that validate previous studies, and provide targets for future translational research into cell type-specific control of fear learning and memory.

## Results

**Generation of Crh-TRAP line**. Crh neurons within the amygdala are found primarily in the CeL (Fig. 1a–c). To examine actively translating mRNA transcripts following FC or EXT, we generated a CRH-TRAP mouse by crossing a Crh-Cre line with a Cre-dependent eGFP-L10a line, which contains a fusion of eGFP with the L10a ribosomal protein (Fig. 1d)[47,57]. Expression patterns of ribosome-tagged eGFP closely recapitulates that observed in native *Crh* expression (Fig. 1e, f).

**Fear conditioniong and extinction in male and female Crh-TRAP mice**. To examine translational changes in Crh neurons following FC and EXT, cohorts of male and female CRH-TRAP mice were generated. Mice were habituated to the behavioral apparatus twice for 10 min, then fear conditioned (FC: 5 CS/US, 30 s conditioned stimulus (CS) at 6 kHz co-terminating with a 0.65 mA unconditioned stimulus (US), 90 s inter-trial interval (ITI)), or exposed only to the TA (TA: 5 × 30 s CS at 6 kHz, 90 s ITI), or fear conditioned then fear extinguished (EXT: 30 × 30 s CS at 6 kHz, 60 s ITI) on consecutive days. Following TA, FC, or EXT, male and female cohorts were sacrificed after a delay of 2 h (Fig. 1g). As expected, FC mice acquired increased freezing responses to tone (Fig. 1h) and EXT mice showed decreased freezing to tone, following the EXT paradigm of 30 CS presentations in the absence of the US (Fig. 1i).

**Differential gene expression analysis**. Differential expression analysis identified differentially expressed genes (DEG) according to the three pairwise comparisons: FC and TA, EXT and TA, and EXT and FC (Volcano plots in Supplementary Fig. 1, DEG lists in Supplementary Data 1). All DEG surviving false discovery rate (FDR) correction were associated with EXT (Fig. 2b) groups, and not with FC (Fig. 2a). Specifically, FDR-significant genes were found in males and not in females, whereas combining both sexes produced a larger set of DEGs (Supplementary Fig. 1), suggesting consistent changes in RNA translation between males and females during EXT, confirmed by correlational analyses (Spearman correlation of effect sizes and rank–rank hypergeometric overlap (RRHO); Supplementary Fig. 2).

The top EXT-associated DEGs (Table 1) were associated with neuronal activity such as the downregulated immediate early genes *Junb* and *Fos*, consistent with decreased neuronal plasticity or potentially LTD-related processes. *Crh* is regulated by glucocorticoids and is central to the HPA axis involved in glucocorticoid function. Notably, genes such as *Crh*, *Dusp1*, *Fkbp4*, *Fkbp5*, *Pja1*, and *Usp22* were differentially expressed and indicated alterations in glucocorticoid receptor signaling.

The correlational analysis revealed significant positive correlations between the FC vs. TA and EXT vs. TA analyses in males, females, and across both sexes (rho = 0.35 – RRHO rho = 0.35, rho = 0.57 – RRHO rho = 0.57, and rho = 0.52 – RRHO rho = 0.52, respectively; Supplementary Fig. 2). This was driven by a

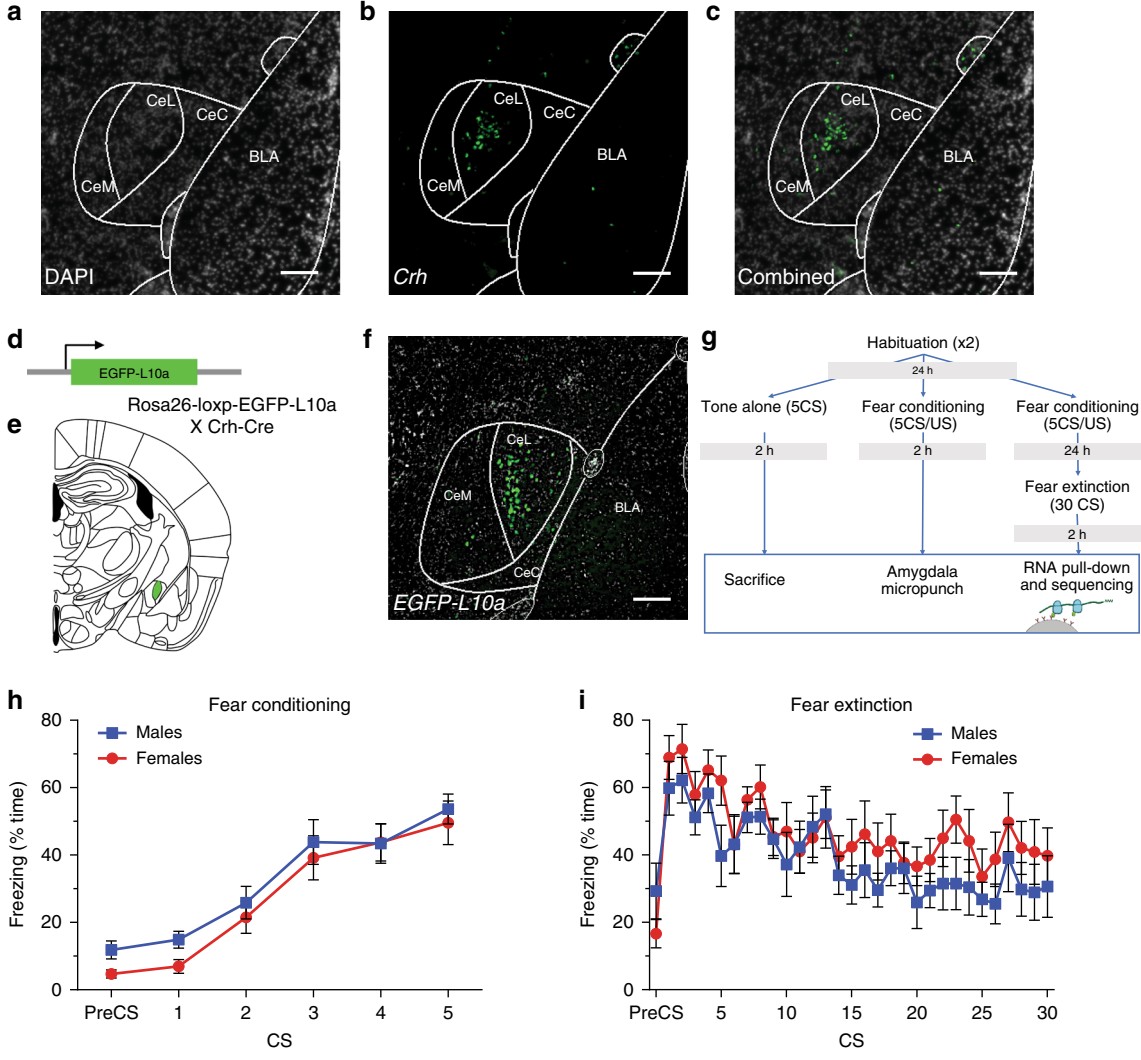

**Fig. 1 Generation of Crh-TRAP line and isolation of CeA Crh cell mRNA after behavior. a–c** In situ hybridization for *Crh* transcripts examining the anterior to posterior axis of the amygdala is presented and has been replicated 28 times using tissue 16 μm slices taken from *n* = 8 mice. **a–c** Representative images: **a** schematic of sub-compartments of CeA (central-CeC, lateral-CeL, and medial-CeM). DAPI (gray). Scale bar: 200 μm. **b**. *Crh* (green) expression is localized largely isolated to a subpopulation of CeL neurons. Scale bar: 200 μm. **c** Overlay of *Crh* with DAPI. Scale bar: 200 μm. **d** Schematic of generation of double transgenic Crh-TRAP mouse. **e** Schematic of *eGFP-L10a* expression in the CeL. **f** Image of *eGFP-L10a* expression within the CeL. Expression of transgene closely recapitulates that of native *Crh* expression. Scale bar: 100 μm. Transgene expression was initially examined in a cohort of mice *n* = 5 with entirely consistent results, of which **f** is a representative image. **g** Schematic of behavioral paradigm and TRAP isolation of *Crh*-specific RNAs. **h** FC of Crh-TRAP animals (*n* = 20 biologically independent animals/group (i.e., sex)). A main effect of CS number was found indicating increased freezing responses with repeated CS/US presentation (two-way RM ANOVA—two-sided: $F(5, 190) = 45.53$, $p = 9.251e{-}31$). No main effect of sex was detected (two-way RM ANOVA—two-sided: $F(1,38) = 0.8492$, $p = 0.98$). **i** EXT of Crh-TRAP animals (*n* = 10 biologically independent animals/group (i.e., sex)). A main effect of CS number was found indicating decreased freezing responses with repeated CS presentation (two-way RM ANOVA: $F(30, 540) = 5.590$, two-sided $p = 6.71e{-}18$). No main effect of sex was detected (two-way RM ANOVA: $F(1,18) = 1.784$, two-sided $p = 0.20$). In **h** and **i**, male and female mice are represented by blue squares and red circles, respectively. Data represented as mean ± S.E.M.

subset of nominal significant genes that are common in both analyses and have the same direction of effect (Supplementary Fig. 3a: 109 downregulated and 56 upregulated shared in males, Supplementary Fig. 3b: 116 downregulated and 147 upregulated shared in females, Fig. 2c: 285 downregulated and 206 upregulated shared in both sexes). However, most of the EXT-associated DEGs were uniquely associated with EXT (in males: 92.79%, in females: 77.95%, and in both sexes: 83.79%).

As noted above, we found many significantly associated DEGs between EXT and TA, but not FC and TA (Fig. 2a, b). The weaker separation between FC and TA groups may be caused by translational changes due to the stress of transport, and handling of animals and the novelty of tone exposure. To test this

hypothesis, a separate cohort of mice was exposed to home cage, TA (5 CS, 30 s CS at 6 kHz, 90 s ITI), or FC (5 CS/US, 30 s CS at 6 kHz, 90 s ITI, 0.65 mA US). Analysis via quantitative PCR (qPCR) of mRNAs isolated from amygdala tissue punches demonstrates that tone exposure is sufficient to initiate a stress-related transcriptional program even in the absence of paired shocks. Furthermore, TA is sufficient to modulate glucocorticoid targets, increase *Crh* and *Sgk1* expression, and decrease expression of *Id3*, as previously reported for FC (Fig. 2d)[58–60].

**Gene network analysis.** Following these validation studies, we applied a number of computational analyses to best understand

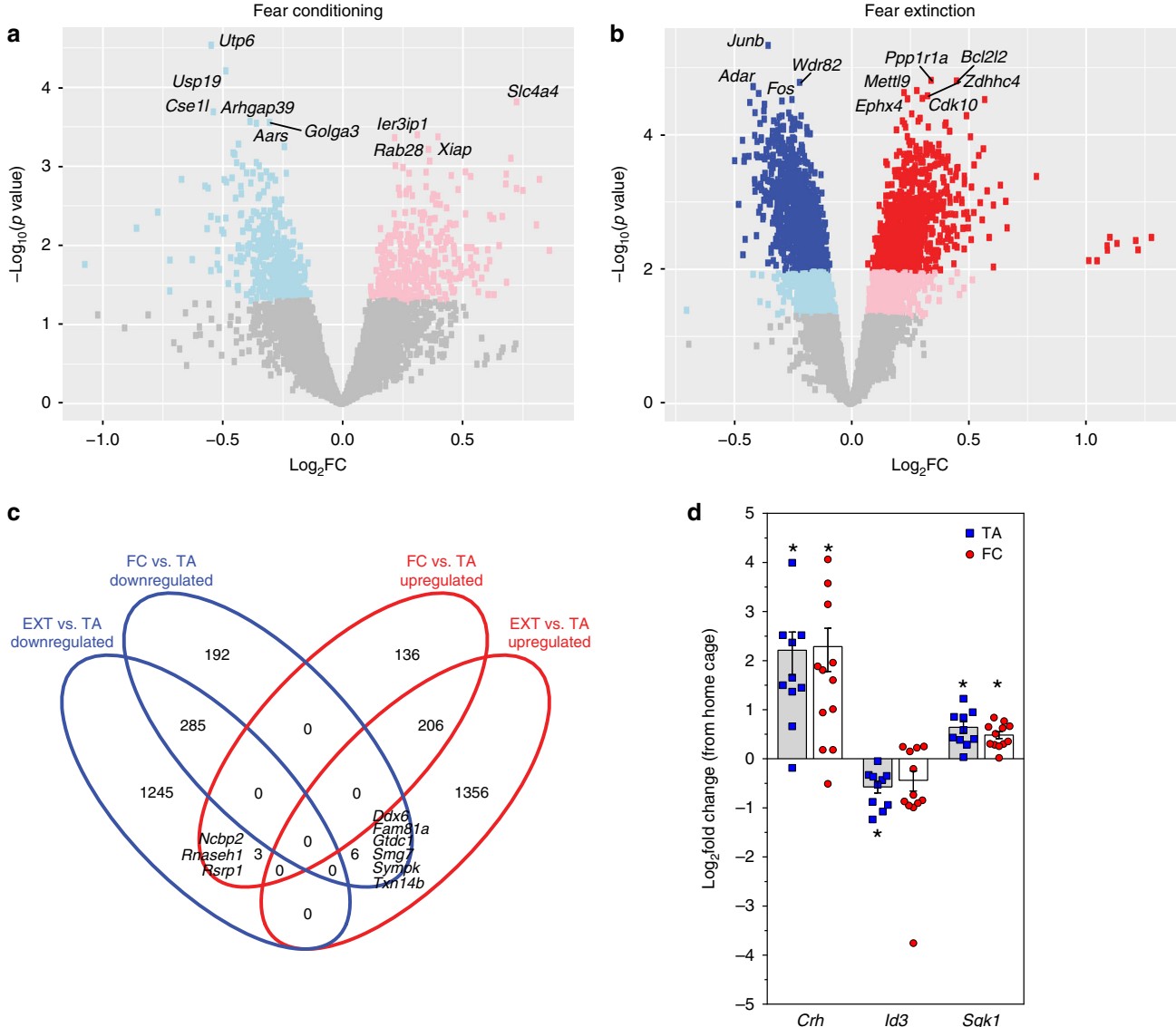

**Fig. 2 Differential gene expression analysis reveals translation signature of Crh cells after EXT. a, b** Volcano plots (i.e., $p$ value (in $-\log_{10}$ scale) by fold changes (in $\log_2$ scale)) of differential gene expression for FC ($n = 12$) vs. TA ($n = 16$) in **a**, and EXT ($n = 16$) vs. TA ($n = 16$) in **b**. Differential gene expression analysis was performed using limma with FDR multiple testing correction of $p$ values. Red squares indicate upregulated genes, while blue indicate downregulated genes. More intense colors indicated FDR-adjusted $p$ value < 0.05. **c** Summarizing the results of differential gene expression analysis, the Venn diagram of upregulated (red ellipse) and downregulated (blue ellipse) DEG by EX or FC compared to TA in both sexes. **d** Quantitative real-time PCR indicated comparable expression differences in *Crh*, *Id3*, and *Sgk1* after TA (blue squares), and FC (red circles) compared to home cage control group ($n = 10$ and 12 biologically independent animals for TA and FC, respectively, $n = 5$ biologically independent animals for the home cage group). $T$ test two-sided $p = 0.012$ and 0.006 for *Crh* upregulation in TA vs. home cage and FC vs. home cage, respectively, $p = 0.009$ for *Id3* downregulation in TA vs. home cage, $p = 0.007$ and 0.009 for *Sgk1* upregulation in TA vs. home cage and FC vs. home cage, respectively). Data represented as mean $\log_2$(fold change from home cage) ± S.E.M. Asterisk indicates $p < 0.05$.

the gene networks that mediate gene regulation in Crh neurons, during the FC and EXT consolidation periods. Network analysis using weighted gene co-expression network analysis (WGCNA) software (Supplementary Data 2) identified 19 gene network modules (noted by different colors) containing 4346 co-expressed genes (Fig. 3a). Based on module eigengene expression (i.e., first principal component of each module expression matrix), six modules significantly associated with EXT, while two modules associated with FC (Fig. 3b; sex-specific analyses in Supplementary Fig. 4a, b) containing a varying number of genes. The Venn diagram in Fig. 3b reveals that five modules were uniquely associated with EXT, containing a variable level of DEGs (Fig. 3c).

**Pathway analysis.** We conducted pathway analyses using GSEA for the DEGs associated genes with FC and with EXT (Supplementary Data 3 and 4, respectively). This analysis identified, as expected, pathways that are primarily uniquely associated with FC (131) and EXT (1579), while 85 shared enrichments showed the same enrichment direction and 7 shared enrichments showed the opposite enrichment direction (Fig. 4a). Top FC-associated enrichments were related with dendritic and postsynaptic gene ontologies (Fig. 4b—GSEA plots in Supplementary Fig. S5a), while top EXT-associated enrichments were related with multiple cellular metabolism and proliferation-related gene networks, further indicating decreased neuronal activity[61], Fig. 4c—GSEA plots in Supplementary Fig. 5b).

**Table 1 The top EXT-associated DEGs identified by TRAP analyses from CeA CRH-expressing neurons.**

| Gene symbol | Gene name | Fold change (log₂ scale) | *p* value | FDR-adjusted *p* value |
|---|---|---|---|---|
| Junb | JunB proto-oncogene, AP-1 transcription factor subunit | −0.351 | 4.65E−06 | 0.012 |
| Ppp1r1a | Protein phosphatase 1 regulatory inhibitor subunit 1 A | 0.343 | 1.54E−05 | 0.012 |
| Bcl2l2 | BCL2 like 2 | 0.452 | 1.57E−05 | 0.012 |
| Wdr82 | WD repeat domain 82 | −0.217 | 1.65E−05 | 0.012 |
| Adar | Adenosine deaminase RNA specific | −0.415 | 1.91E−05 | 0.012 |
| Mettl9 | Methyltransferase like 9 | 0.282 | 2.19E−05 | 0.012 |
| Ephx4 | Epoxide hydrolase 4 | 0.228 | 2.35E−05 | 0.012 |
| Fos | Fos proto-oncogene, AP-1 transcription factor subunit | −0.391 | 2.41E−05 | 0.012 |
| Zdhhc4 | Zinc finger DHHC-type palmitoyltransferase 4 | 0.328 | 2.62E−05 | 0.012 |
| Cdk10 | Cyclin-dependent kinase 10 | 0.306 | 2.86E−05 | 0.012 |
| R3hdm4 | R3H domain containing 4 | 0.243 | 2.90E−05 | 0.012 |
| Camk2n1 | Calcium/calmodulin-dependent protein kinase II inhibitor 1 | 0.572 | 2.98E−05 | 0.012 |
| Prpf18 | Pre-mRNA processing factor 18 | −0.250 | 2.98E−05 | 0.012 |
| Dhx30 | DExH-box helicase 30 | −0.293 | 3.13E−05 | 0.012 |
| Pja1 | Praja ring finger ubiquitin ligase 1 | −0.430 | 3.30E−05 | 0.012 |
| Dlg3 | Discs large MAGUK scaffold protein 3 | −0.367 | 3.51E−05 | 0.012 |
| Usp22 | Ubiquitin-specific peptidase 22 | −0.369 | 4.38E−05 | 0.012 |
| Usp19 | Ubiquitin Specific Peptidase 19 | −0.407 | 4.68E−05 | 0.012 |
| Ubash3b | Ubiquitin associated and SH3 domain containing B | −0.263 | 4.73E−05 | 0.012 |
| Fam120a | Family with sequence similarity 120A | 0.261 | 4.87E−05 | 0.012 |

Top 20 genes from differential gene expression analysis for EXT ($n = 16$) vs. TA ($n = 16$), which was performed using limma with FDR multiple testing correction of *p* values.

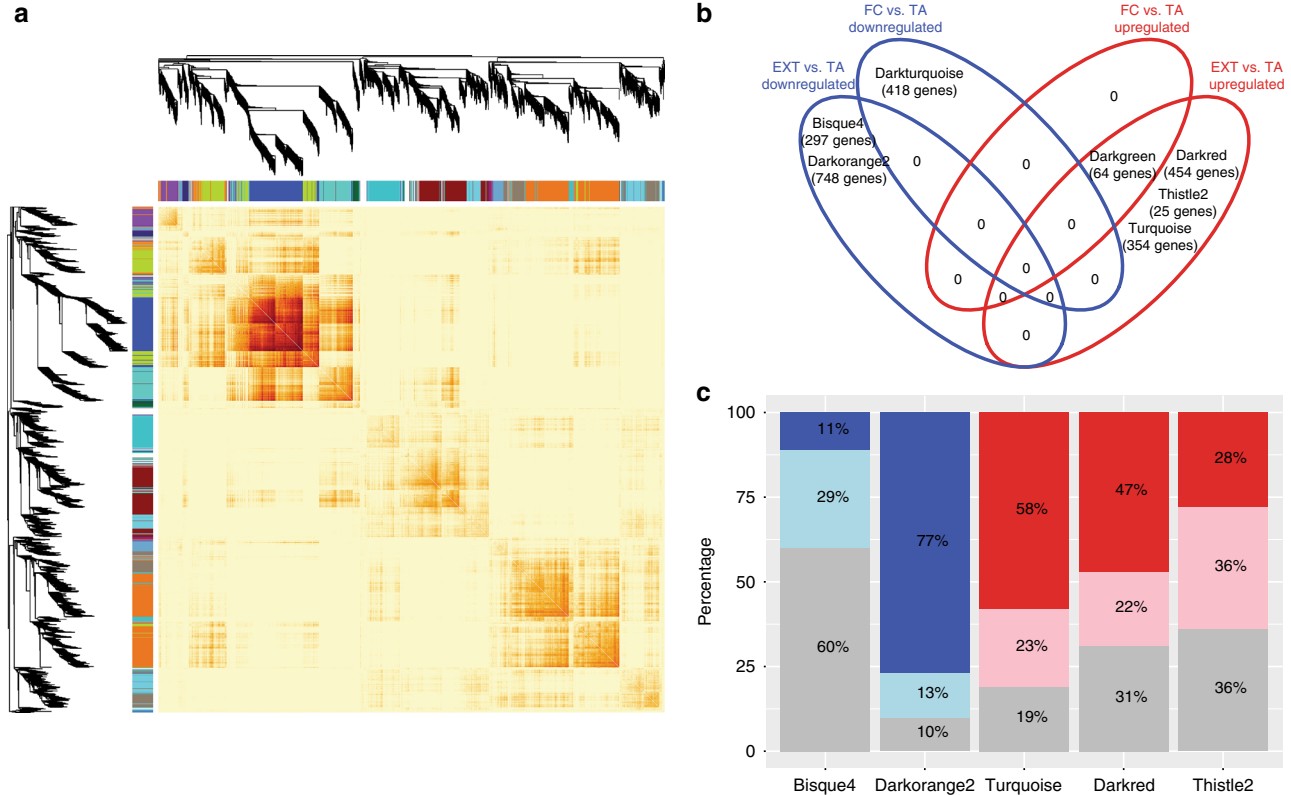

**Fig. 3 WGCNA identifies gene networks associated with EXT. a** Heatmap plot depicting the co-expression topological overlap matrix (interaction patterns among genes using correlation as a measure of co-expression), supplemented by hierarchical clustering dendrograms, and the module colors for the weighted gene co-expression network analysis (WGCNA) analysis. Please note network names are colors by convention. **b** Network module eigengenes were used for differential expression for FC ($n = 12$) vs. TA ($n = 16$), and EXT ($n = 16$) vs. TA ($n = 16$) using limma with FDR multiple testing correction of *p* values. Summarizing the results of differential module eigengene expression analysis, the Venn diagram revealed unique and shared upregulated (red ellipse) and downregulated (blue ellipse) gene network modules by FC or EXT compared to TA in both sexes, **c** The barplot depicted the distributions of genes included in uniquely EXT-associated networks ("bisque4", "darkorange2", "turquoise", "darkred", and "thistle2") into EXT-associated DEGs (blue or red) or not (gray). Red indicates upregulated genes, and blue indicates downregulated genes. More intense color indicates FDR-adjusted *p* value < 0.05.

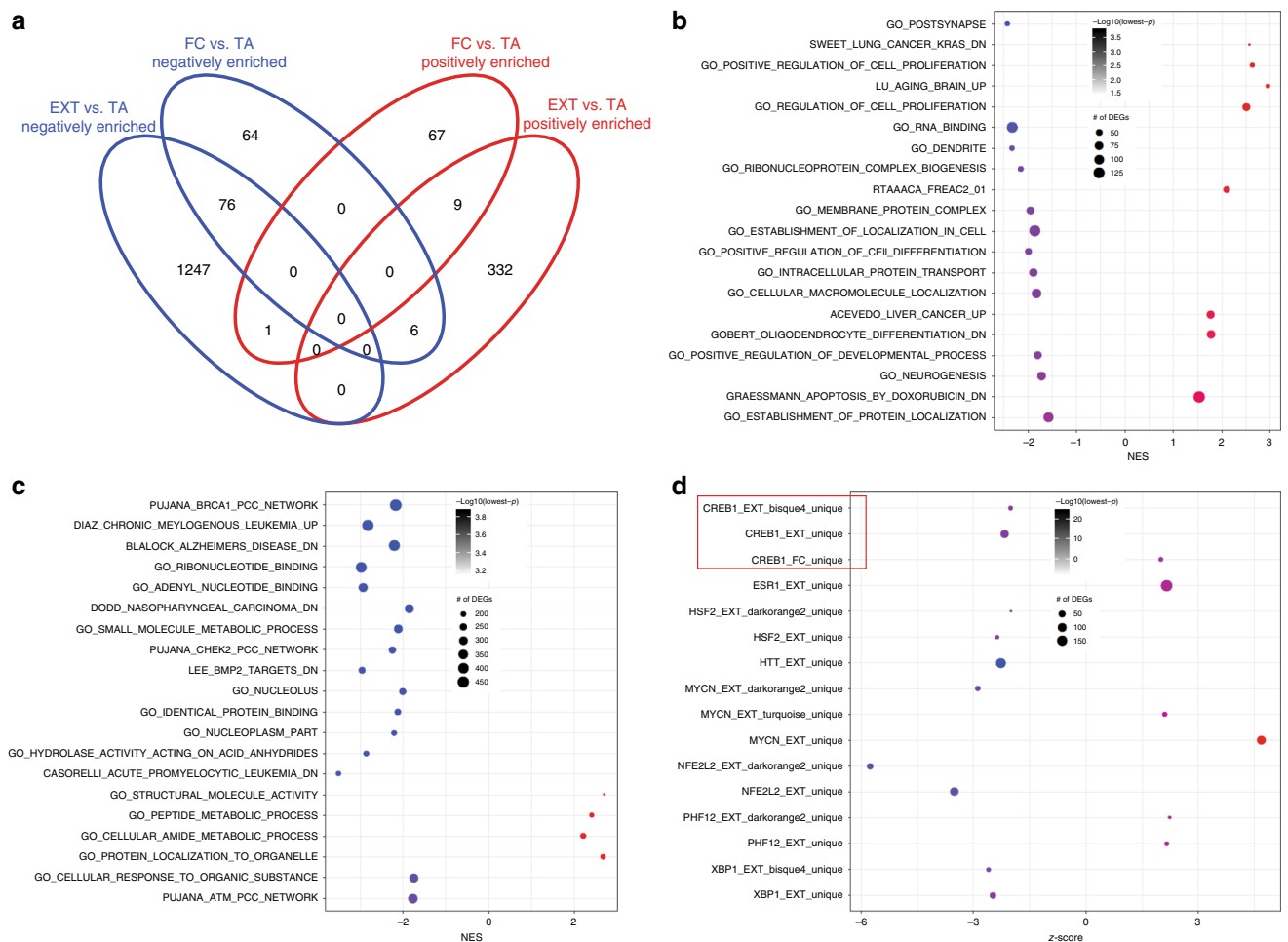

**Fig. 4 GSEA and URA of FC- and EXT-associated differential expression across sexes. a–c** GSEA was used to test concordance of differential gene expression analyses results with gene expression signatures from the Molecular Signatures Database. Enrichment scores are calculated against background and they are also corrected for multiple comparisons based on FDR. Summarizing the GSEA results, the Venn diagram revealed unique, and shared positively (red ellipse) and negatively enriched (blue ellipse) gene sets in FC or EXT compared to TA in both sexes (**a**), while the bubble plots depict the uniquely FC-enriched gene sets (**b**), and uniquely EXT-enriched gene sets (**c**) with the lowest *p* value (top 20). Gene sets with negative normalized enrichment score (NES) are depicted with blue circles, while those with positive NES with red circles. The size of the circles corresponds to number of underlying DEGs. **d** DEGs together with their fold changes were used for the prediction of activation/deactivation of upstream regulator by URA based on Fisher's exact test *p* value and activation *z*-score. Bubble plot for the significant FC- or EXT-associated upstream regulators based on DEGs or genes of differentially expressed modules uniquely associated with FC and EXT, respectively. Upstream regulators with negative *z*-score are depicted with blue circles, while those with positive *z*-score with red circles. The size of the circles corresponds to number of underlying DEGs. The predictions of CREB1 activity are highlighted with a red box.

**Upstream regulator analysis**. We then conducted URA of genes uniquely associated with FC or EXT, or included in modules uniquely associated with FC or EXT (Fig. 4d and Supplementary Data 5). Regulation of DEGs and differentially expressed networks revealed upstream regulators. CREB1 stood out in this analysis as it was predicted to be activated based on FC DEGs, and deactivated based on EXT DEGs and bisque4 EXT DEGs. Importantly, bisque4 was also the *Crh* gene-containing network. Together these findings suggested that genes mediated by the CREB pathway are integral to differential regulation of FC and EXT within the Crh-specific cell population in CeA.

**CREB overexpression effects in behavior**. To validate and further characterize the role of Crh neuronal CREB in the expression and EXT of fear, we microinfused AAV vectors containing a cre-recombinase (Cre)-sensitive DIO construct encoding CREB into the CeA of mice expressing Cre in Crh neurons (CRH-Cre),

thereby inducing elevated expression of CREB within the Crh neurons in this region (CeL; Fig. 1b). Three weeks after gene transfer, mice were tested in a battery of behavioral tests, ending with FC and EXT (Fig. 5a). Histological analysis examining the GFP reporter protein from the CREB and control viral constructs indicated strong transgene expression in the CeA that is consistent with patterns of native *Crh* expression, which is restricted to the CeL (Fig. 5b, c). No differences between groups were found in anxiety-like behavior in the elevated plus maze or the open field test (Fig. 5d and e, respectively), or in overall levels of locomotor activity (Fig. 5f).

Subsequently, mice were exposed to a mild FC regimen (5 CS/US, 0.5 mA, 0.5 s US) that typically produces subthreshold changes in freezing behavior. As expected, no within-session differences in FC were evident during the training (Fig. 5g). However, when mice were tested for fear expression to the tone 24-h later, those with viral vector-induced elevations in CREB in CeA Crh neurons showed enhanced freezing throughout the

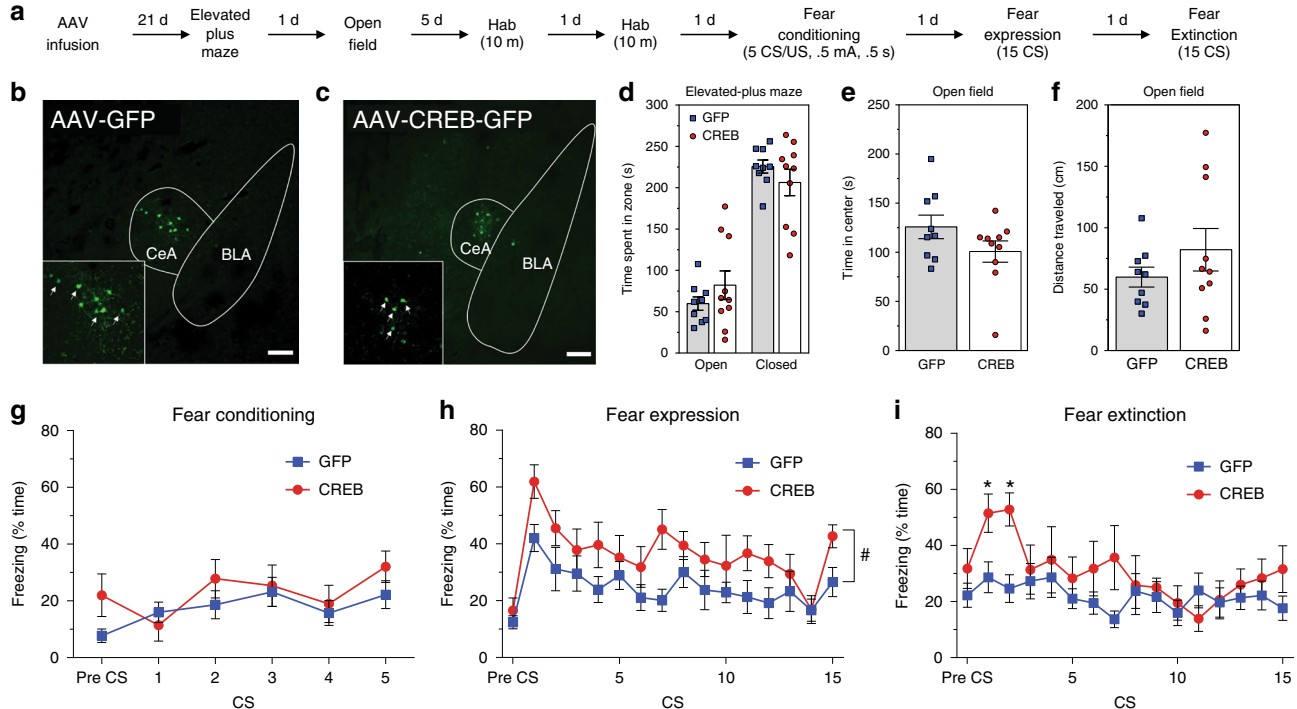

**Fig. 5 Overexpression of CREB in Crh cells enhances fear expression and blunts EXT. a** Experimental workflow. **b, c** Infusion of control AAV-DIO-GFP (**b**) and experimental AAV-DIO-CREB-GFP (**c**) viruses into the amygdala leads to transgene expression specifically in CeA Crh neurons and, sporadically, within BLA. Scale bar in **b, c**: 100 μm. Viral expression was examined for each GFP and CREB animal. Behavioral data from those with significant reporter expression, as depicted in **b, c**, were retained and used to produce (**d-i**). One GFP mouse was discarded for lack of expression leaving $n = 9$ GFP and $n = 10$ CREB mice. **d, e** Overexpression of CREB in Crh neurons does not lead to significant changes in anxiety as measured by: **d** time spent in open arm of an elevated plus maze (t test two-sided $p = 0.27$, $n = 9$ and 10 biologically independent animals/group for GFP and CREB respectively); **e** time spent in the center of an open field (t test two-sided $p = 0.14$, $n = 9$ and 10 biologically independent animals/group for GFP and CREB, respectively). **f** No changes in locomotion were detected in open field (t test two-sided $p = 0.28$, $n = 9$ and 10 biologically independent animals/group for GFP and CREB, respectively). **g** No differences were found with CREB overexpression during FC ($n = 9$ and 6 biologically independent animals for GFP and CREB, respectively; two-way RM ANOVA: $F(1,13) = 2.120$, two-sided $p = 0.16$). **h** CREB overexpression in Crh neurons increases fear expression ($n = 9$ and 6 biologically independent animals for GFP and CREB, respectively; two-way RM ANOVA: $F(1,13) = 8.244$, two-sided $p = 0.013$). **i** CREB overexpression leads to a nonsignificant increase in fear expression during first CSs of EXT retention test ($n = 9$ and 6 biologically independent animals for GFP and CREB, respectively; two-way RM ANOVA: $F(1,13) = 3.748$, two-sided $p = 0.0749$; post hoc t test for individual CS significance, two-sided $p = 0.021$ and $p = 0.0029$ for CS1 and CS2, respectively). Data represents mean ± S.E.M. In **d-i**, GFP and CREB mice are represented by blue squares and red circles, respectively. Hash sign (#) in **h** indicates an overall group difference ($p < 0.05$), while asterisk (*) in **i** indicates differences in particular CS trials ($p < 0.05$).

session (Fig. 5h). In addition, when tested 24 h following EXT, CREB overexpressing mice again expressed more freezing during initial tone presentations. Together these data suggest that enhanced CREB expression in CeA Crh neurons promotes fear expression and may blunt EXT. These findings also complement the cell type-specific RNAseq observation that reductions in *Creb* within CeL Crh neurons accompany EXT by demonstrating that offsetting this change facilitates fear expression and opposes EXT.

## Discussion

We used TRAP-seq to isolate Crh neuron-specific ribosome-associated RNAs following tone alone exposure (TA), fear conditioning (FC), or extinction (EXT). Bioinformatic analysis of translational changes in Crh neurons following EXT identified differentially regulated genes and gene pathways involved in this learning process. Furthermore, functional validation of one of these gene pathways, confirmed the TRAP-based findings of CREB regulation underlying EXT of conditioned fear.

Notably, we did not observe FDR-significant differences between TA and FC. Because Crh neurons are recruited during learning of weak threats and active responses, we hypothesized that stress derived from a combination of events ranging from

transport, handling, and tone exposures was likely sufficient to activate this neuron population, washing out differences between TA and FC groups[40]. This hypothesis was tested in a separate cohort of mice demonstrating that stress-related genes *Crh*, *Sgk1*, and *Id3* are regulated in response to both TA and FC. Thus, it is likely that the translational signature of associative fear learning was obscured by more generalized stress-related translational changes.

It is important to also note that a population marked by expression of a gene is not synonymous with continuous high levels of expression of that gene and should be considered more as an indicator of shared origin or function. Thus, *Crh* expression within the Crh neuron population is dynamic and, while the transgenic mouse line used reliably captures the Crh neuron population, *Crh* expression may vary between cells depending on behavioral state[62]. Regardless, our data demonstrate gene changes as a result of EXT, and indicate the transition from an activated stress translational program to a deactivated stress translational program.

Significant DEGs were identified between EXT and TA in males and across sexes. Changes in expression between males and females were strongly correlated; however, no FDR-significant DEGs were identified in the female group alone. This may be due to inclusion of females without regard to estrus cycle phase and

phasic alterations in gene expression. Estrus cycle phase is known to alter gene expression in the amygdala, synaptic plasticity, and EXT[63–65].

DEG's associated with EXT identified changes in translation of several activity and stress-related transcripts. These changes are consistent with decreased activity and suppressed stress hormone responses in Crh neurons during EXT. The data are also consistent with previous reports of decreased Crh neuronal activity and reductions in stress-related gene activity with EXT[17,40,42,43]. Analysis of regulation of EXT genes revealed substantial overlap with genes known to be regulated in neuropsychiatric disease. Notably, the CRH receptor (CRHR1) has now been associated with PTSD symptoms, anxiety disorders, and alcohol habitual use in large-scale GWAS studies[11–14]. In addition, prior work in GWAS association of pathways across psychiatric disorders has demonstrated that the protein encoded by CREB1 directly interacted with several risk genes of psychiatric disorders identified by GWAS[66].

While genes are co-expressed forming functional networks, understanding specific aspects of these genes and networks can provide insight into cellular function. In our study, the majority of EXT-related DEGs were co-expressed in five networks. Regulation of DEGs and differentially expressed networks revealed a list of upstream regulators. Further understanding of these upstream regulators may provide critical insight into the role of Crh neuronal populations in the consolidation of fear EXT memories.

Previous work has shown that cell type-specific modulation of specific genes dramatically affects FC and EXT[67–69]. To validate our upstream regulator analyses, CREB was chosen for cell type-specific manipulation given that it was affecting the gene network containing Crh, and it has been shown to be necessary for stress-related increases in Crh transcription. Although modulation of CREB has been associated with other amygdala- and striatal-dependent memory processes[53,55,70,71], its function has never been examined specifically within the CeA Crh neuronal population. Congruent with predictions from our TRAP-seq data, overexpression of CREB in Crh neurons enhanced fear expression and may blunt EXT. Together these findings suggest that CREB within a specific subset of neurons (Crh), embedded within a specific subregion of amygdala (CeL), functions as a molecular switch that regulates expression of fear. More broadly, they also confirm the general principle that CREB is involved in producing behavioral outcomes that range from being therapeutic to maladaptive, depending on the specific brain region and cellular subtypes being affected, complicating the development of medications that would nonspecifically produce activation or inhibition of its function[53,72–74].

The CREB construct used in cell type-specific overexpression experiments has been used in a wide variety of experiments spanning decades by the Carlezon and other labs. The Carlezon group has demonstrated that overexpression of CREB leads to increases in CREB-mediated gene transcription and CREB-mediated changes in electrophysiological responses. In addition, they have demonstrated that CREB overexpression mimics pCREB-like activation of gene expression[52,75,76]. Given that Pavlovian FC is explicitly an associative learning assay, overexpression of CREB in Crh neurons would not be predicted to nonspecifically increase freezing at baseline in the absence of a learned association; this prediction was supported by our results. Overall, these data demonstrate that cell type-specific analyses of translational gene regulation is robust, identifying both expected and previously not appreciated pathways, and pointing to targets for manipulating fear expression and extinction.

A limitation of these studies is that our experiments of CREB expression in CeA Crh neurons were completed only in male mice. In female mice, estrus cycle and sex-specific effects may have important roles, and will be addressed in future studies. Another limitation relates to our interpretation of CREB overexpression effects on enhanced fear expression vs. blunted fear extinction. Given that CREB is overexpressed in the experimental group during both fear expression and EXT sessions, it is possible to examine if overexpression of CREB in Crh neurons enhances fear expression; however, experimental parameters do not differentiate whether greater freezing during EXT session is solely due to enhanced fear expression or deficits in EXT.

This work adds to our understanding of the role of the Crh amygdala neurons, and the increasingly appreciated importance of the Crh regulatory pathway in trauma and stress-related disorders, such as PTSD. Cell type-specific targeting of CREB or other fear extinction related genes for knockdown, inhibition, or activation may reveal translationally relevant pathways for intervening in fear-related pathologies. This type of comprehensive, yet cellularly precise, analysis offers a potentially important array of targets that may be useful for the diagnosis, treatment, and prevention of psychiatric illness.

In conclusion, we examined differential gene expression specifically within the amygdala Crh neuronal population, comprised primarily of the CeA Crh-expressing neurons. Our analyses revealed that translational profiles after EXT learning were consistent with overall decreased neuronal activity in these neurons. Gene co-expression network analysis identified gene networks activated or inhibited by EXT learning, and URA identified CREB as a critical pathway downregulated with EXT. Finally, we confirmed that overexpression of CREB in CeA Crh neurons increased fear expression and may blunt fear extinction, as predicted from the TRAP-seq data.

## Methods

**Animals**. All mouse lines were obtained from Jackson Laboratories (Bar Harbor, ME). For all experiments, mice were between 10–16 weeks old at the time of behavioral training and sacrifice. For generation of Crh-TRAP line, a Crh-Cre line (B6;FVB-Tg(Crh-cre)1Kres/J) was crossed with a cre-dependent TRAP reporter line (Rosa26 fs-TRAP; (B6.129S4-Gt(ROSA)26Sortm1(CAG-EGFP/Rpl10a,-birA) Wtp/J); referred to for short as the "eGFP-L10a line"). Only first-generation progeny was used for TRAP experiments ensuring that mice were heterozygous for each transgene. For qPCR validation of gene expression changes following TA and FC, C57BL/6J mice were used. For follow-up validation of Crh neuron-specific overexpression of CREB, B6(Cg)-Crhtm1(cre)Zjh/J were crossed with wild-type C57BL/6J breeding partners, and only first-generation progeny were utilized ensuring all mice were heterozygous for the transgene. Both Crh-cre lines have previously been validated to have accurate targeting of Cre expression to Crh neurons in the CeA[57,69,77]. Mice were maintained on a standard 12 h light:12 h dark light cycle. All mice were group housed with 2–5 same-sex litter mates. Mice were housed in a temperature and humidity-controlled facility, and given free access to food and water. All procedures were approved by McLean Hospital Institutional Animal Care and Use Committee, and complied with National institutes of Health guidelines. For behaviors, animal numbers were informed by previous experiments to inform power and effect size calculations, using the G*Power 3 software package. Animals were all randomized and assigned to behavioral or viral manipulation groups by an experimenter blinded to the experimental conditions.

**Viral-mediated gene transfer**. Plasmids for Cre-dependent AAV (HAR-EF1a-DIO-CREB1B-IRES-GFP-SV40pA and HAR-EF1a-DIO- GFP-SV40pA) were generated by Dr. Rachael Neve (Massachusetts General Hospital Gene Transfer Core), and packaged into an AAV 8.2 capsid by Virovek Inc.

For viral manipulation experiments, mice were anesthetized deeply with a ketamine/xylazine mixture prior having their heads fixed into a stereotaxic apparatus (Kopf). During surgery, body temperature was maintained using a heating pad. Stereotaxic coordinates (A/P −1.4, M/L ±2.9, D/V −4.4) were taken from Sanford et al.[40] and confirmed in Paxinos and Franklin[78]. Heads were leveled, and virus was delivered bilaterally through burr hole in skull via a 1.0 μl microsyringe (Hamilton). Syringe was lowered to coordinates and 0.3 μl of virus was infused at a rate of 0.1 μl/min followed by a 12-min resting period. Following infusion, syringe was slowly withdrawn over 5 min. Following infusion, incision was closed by suturing with a nylon monofilament (Ethicon). Mice were allowed to recover and regain mobility.

**Behavioral assays**. For all FC and EXT experiments involving gene analyses, mice were habituated to the chamber (Med Associates) for 10 min for the 2 days preceding FC. On the day of training, mice were exposed to five tone-shock pairings (pre-CS period: 180 s, CS: 30 s 6000 Hz, 65–70 db, co-terminating shock (US): 0.5 s, 0.65 mA, ITI between CS's: 90 s). For the TA control group, shocks were omitted by turning shock generator off. The EXT group was returned to the animal care facility, while FC and TA groups were left in holding room until sacrifice. The next day EXT group was tested in a novel context. Alternative context was provided in different set of apparatuses in a different room and had different olfactory cue, lighting conditions, and flooring. Mice were exposed to 30 CSs in the absence of any US reinforcer (pre-CS period: 180 s, CS: 30 s 6000 Hz, 65–70 db, ITI between CS's: 60 s). Following EXT, mice were returned to the holding room until sacrifice. Freezing was measured using FreezeFrame software (Coulbourn Instruments). For viral infusion experiments, the same method was used except the US was 0.5 mA and only 15 CSs were presented during extinction.

For the open field test, mice were placed in a 44 cm cubed box in a dimly lit room. Behavior was recorded for 10 min, while mice were allowed to explore. Distance traveled and time spent in center area were calculated using Ethovision software (Noldus).

For the elevated plus maze test, animals were placed into the center of an elevated plus maze facing open arm. Apparatus had arms measuring 50 cm tip to tip and was placed in a dimly lit room. Mice were allowed to freely explore for 10 min, while behavior was recorded. Time spent in open arms, closed arms, and center as calculated using Ethovision software.

**Fluorescent in situ hybridization–RNAscope staining**. In situ hybridization to localize *Crh* transcripts was performed on sections taken from eight adult male C57BL/6J mice. Mice were briefly anesthetized with isofluorane, decapitated, brains removed, and snap-frozen. Slices were taken at a width of 16 μm. RNA scope procedure was performed to manufacturers specifications (ACD Bioscience) using mm-Crh-C1 probe and RNA Scope Fluorescent Multiplex 2.5 labeling kit (ACD Bio).

**Image acquisition**. Images of in situ staining, transgene expression and viral reporter expression were acquired on a Leica SPS confocal microscope using a 10× or 40× objective. Images were acquired using z-stacks with computer optimized step size. Max intensity projections were generated and presented. Image signals were quantified using ImageJ software.

**Translating ribosome affinity purification RNAseq analysis**. TRAP was performed in accordance with methods published by Heiman et al.[47]. Adult Crh-TRAP mice were quickly anesthetized with isofluorane, decapitated, brains removed, and snap-frozen. Punches centered over the amygdala were taken bilaterally using a 1 mm punch. Individual animals were used as each sample (n = 10 per condition/sex). Tissue was homogenized and mRNA's isolated from GFP-tagged ribosomes (TRAP). RNA quantity and quality were assessed using Bioanalyzer Pico Chip (Agilent).

Libraries were prepared using SMARTer HV kit (Clontech) and NexteraXT DNAkit. Microelectrophoresis was used to validate libraries followed by quantification, pooling, and clustering on the Illumina TruSeq v3 flowcell. Clustered flowcell was sequenced using an Illumina HiSeq 1000 in 50-bp paired end mode. Twenty-five million reads per sample were targeted.

**RNAseq data processing**. Sequence reads were trimmed to remove possible adapter sequences and nucleotides with poor quality using Trimmomatic v.0.36. The reads were then mapped to the *Mus musculus* GRCm38 reference genome available on ENSEMBL using the STAR aligner[79]. There resulting BAM files were sorted using the samtools sort function and used for gene counting. Count files were processed with edgeR[80]. A total of 8204 gene symbols with cpm >4 in all samples were retained. Samples were retained according to the full capturing of their behavioral data and based on outlier determination, according to a consensus of hierarchical clustering, PCA and MDS. A total of 44 samples were retained. Log-transformed data were then voom normalized[81].

**Real-time PCR**. RNA from amygdala punches was isolated using Qiagen RNeasy mini kit as indicated by manufacturer. Real-time PCR was run on cDNA from each sample in triplicate. Reactions were run containing 5 μl SYBR, 0.5 μl each forward and reverse primers, 3 μl water, and 1 μl cDNA. Real-time PCR was run on Applied biosystems 7500 Real-time PCR System with cycling parameters of: 10 min at 95 °C, 40 cycles of 15 s at 95 °C, and 60 s at 60 °C. *Itm2b* was used as housekeeping gene. 2^-ddCT values were calculated to represent expression as fold change compared to home cage ± S.E.M.

**Differential gene expression**. Differential expression analysis was performed with limma[82] controlling for experimental batch and sex, when the analyses included both sexes. We used the default multiple testing correction of $p$ values in limma, which is Benjamini–Hochberg's FDR. To compare results from different genome-

wide analyses, we calculated Spearman rank correlations (rho) of the fold changes. RRHO analyses were performed to identify significant overlap of differential expression lists between pairs of results by determining the degree of statistical enrichment using the hypergeometric distribution[83].

**Gene network analysis (WGCNA)**. WGCNA was performed at the default setting using all the samples to identify co-expression networks (modules). Module eigengenes were used for differential module expression using limma.

**Gene set enrichment analysis**. GSEA implementation was done in R[84], *fgsea*, was used to test concordance of differential gene expression analyses results with gene expression signatures from the Molecular Signatures Database (MSigDB, datasets: http://software.broadinstitute.org/gsea/msigdb). Fgsea uses as input file: (a) the ranked DEG list ($s_i$ = sign(fold change gene $i$) × ($-\log_{10}(P_i)$)), and (b) a list of pathways sets. Fgsea in order to calculate the $p$ values for the pathways: (a) calculate a local gene-level statistic, (b) calculate a global gene set level statistic, (c) determine significance of the global statistic (calibration against the background distribution using permutation test), and (d) adjust for multiple testing. The final $p$ value is the fraction of the permutation null values greater than or equal to the observed one.

URA[51] was performed as we described before[85] to estimate a set of significant (Fisher's exact test $p$ value < 0.05) deactivated ($z$-score < −2) or activated ($z$-score > 2) upstream regulators (i.e., transcription regulator or ligand-binding nuclear receptor). URA was performed for (i) the uniquely FC- or EXT-associated DEGs with $p$ value < 0.05 and (ii) the genes belonging to the uniquely FC or EXT co-expressed modules with significant FC or EX differential eigengene expression, respectively.

**Statistics**. Statistical analyses were performed using GraphPad Prism 7 and 8. All data are represented as a mean ± S.E.M. Statistical significance was set at $p <$ 0.05. Freezing during FC and EXT experiments was analyzed using a repeated-measures ANOVA, with group (sex in Fig. 1h, i and virus type in Fig. 5g–i) as between subject factor and tone presentation as the within subject factor. In the case of the test completed in Fig. 5i, a trending (trends considered to be $p < 0.1$) ANOVA ($p = 0.07$) was identified and post-hoc $t$ tests were completed. Behavioral measures in the open field or elevated plus maze (Fig. 5d–f) of the two virus types were compared using a Student's $t$ test. For qPCR (Fig. 2d), expression data from TA and FC groups were compared with home cage group by Student's $t$ test.

**Reporting summary**. Further information on research design is available in the Nature Research Reporting Summary linked to this article.

## Data availability

The raw and processed TRAP-seq datasets generated during the current study were deposited as GEO accession number GSE157021. Source data are provided with this paper.

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

## Acknowledgements

This work was supported by funding from Cohen Veteran Biosciences, NIMH (P50-MH115874, R01-MH108665, and R01-MH117292), and the Frazier Institute at McLean Hospital. C.C. was supported by the 2019 Seed Grant (through NIMH P50-MH115874). N.P.D. was supported by a NARSAD Young Investigator grant and an appointed KL2 award from Harvard Catalyst/The Harvard Clinical and Translational Science Center (NCATS KL2TR002542 and UL1TR002541).

## Author contributions

K.M.M., N.P.D., and K.J.R. conceived and designed the study. Funding was obtained by N.P.D. and K.J.R. K.M.M. performed primary behavioral experiments, viral infusions, and histological analyses, while J.H., G.M., and R.J.F. provided technical assistance. R.L.N. prepared the viral vector. N.P.D. designed and performed primary TRAP-seq data analysis. N.P.D. and C.C. performed the downstream bioinformatic analysis. Additional feedback and discussion were provided by W.A.C.. K.M.M., N.P.D., and K.J.R. wrote the paper with input from all the authors. C.C. and J.H. contributed equally.

## Competing interests

Within the past 2 years, W.A.C. has served as a paid consultation for PSY Therapeutics for unrelated work. N.P.D. has held a part-time paid position at Cohen Veteran Biosciences, has served as a paid consultant for Sunovion Pharmaceuticals and is on the scientific advisory board for Sentio Solutions, Inc. for unrelated work. K.J.R. has received consulting income from Alkermes, and is on scientific advisory boards for Janssen, Verily, and Resilience Therapeutics for unrelated work. He has also received sponsored research support from Takeda and Brainsway for unrelated work. All other authors declare no competing interests.
