## [Peer Review File · Nature Communications]

Reviewers' comments:

Reviewer #1 (Remarks to the Author):

Crh-expressing neurons are found with the CeL of the CeA and have been implicated in fear expression and extinction. This paper uses TRAP-seq to investigate how translation changes in Crh neurons in the CeL following fear conditioning and extinction. DEGs following extinction, compared to tone exposure only, suggest a decrease in neuronal activity or plasticity and altered glucocorticoid receptor signaling. Interestingly, fear conditioning compared to tone only did not produce many DEGs, which appeared to be due to stress-related translational changes associated with behavioral testing itself. Upstream regulator analysis indicated CREB as a potential regulator of fear expression and extinction. This finding was validated with viral overexpression of CREB, which was sufficient to increase fear expression. Overall this is a very good paper with a few major issues. It is important to verify a few changes before its accepted: more support for explaining away sex differences in DEGs after extinction and a re-interpretation of their extinction behavior in their key validation experiment since it appears that the authors may be over-stating their results significantly.

Major concerns:

Since males have significant DEGs, but females do not, the authors should consider an additional approach to help visualize the DEGs in females versus males after EXT. The correlations in the supplement are helpful to show that the sexes appear to be changing in a similar way, but an RRHO analysis or even a venn diagram comparing EXT v. TA in males and females would be an easier, more digestible, way to show that genes are being similarly up or down regulated in males and females. Alternatively, the authors could use a slightly less stringent cut-off and describe these genes as not surviving FDR correction, but very close to reaching significance. This is a critical point since DEGs are not significant in females, that could suggest these genes are less important functionally or more variable. This brings up an important second point, these genes might be more variable in females due to cycling sex hormones. Did the authors verify estrous cycle phase in these female mice? Perhaps they were sacrificed at different phases leading to increased variability, making it more difficult to survive FDR correction.

In the site-specific viral experiments sex is not discussed. Were both sexes used in this experiment? Since upstream analysis appears to have been conducted across sex it could be very important to confirm overexpression of CREB increases fear expression in both males and females independently.

Furthermore, the suggestion that CREB overexpression blunts extinction learning is too strong. Subthreshold fear conditioning is used here in order to show increased fear expression in CREB overexpressing mice, but that results in a GFP group that doesn't show any fear extinction. CREB mice are showing fear expression (higher CS1 compared to preCS) and subsequent extinction (higher on CS1 and lower at CS15), but the GFP mice are not showing either. Therefore, extinction in CREB overexpressing mice artificially appears to be blunted. This "blunted" extinction is just a carry-over from these CREB mice showing fear expression in the first place when GFP controls do not and therefore have nothing to extinguish. This interpretation should be re-stated or an experiment should be added to confirm that extinction is in fact blunted under normal fear conditioning parameters, when control mice would show fear expression and extinction.

Minor concerns:

The authors should be sure to carefully check for typos and punctuation issues. There are several places where words appear to be missing.

In figure 1, the authors should consider removing panel b since it doesn't add very much to the figure and there is no non-DAPI comparison image included for the transgene.

The timeline (g) is also very confusing and makes it seem as though the EXT group were only exposed to tones. The authors should re-work this flowchart to make it clear that the EXT group was fear conditioned. For example, right justify or and tone alone and extend the arrow so it goes from FC to EXT.

This caption also says gender instead of sex, which should be used in this case and is used in the rest of the manuscript.

In figure 3B it appears dark green is in the wrong place. Based on the text, it appears dark green should be a network that is overlapping between FC v TA and EXT v TA. That cell is also empty.

Captions for Suppl. Fig 3 and 4 are the same.

To what extent is CREB overexpressed?

In the site-specific viral experiment:

Sample sizes are not given for these behavioral experiments nor are ns statistics. If stats are given for ns fear conditioning, they should be given for EPC etc. Especially given the small differences that can be seen in these anxiety-like behaviors, like a decrease in center time in open field in CREB mice.

The authors should also discuss why they did not expect to find freezing in CREB overexpressing mice during this subthreshold fear conditioning protocol. If CREB mediates fear expression, why wouldn't freezing be increased during conditioning, as well as in the fear expression test.

The authors should also discuss whether CREB knockdown might be therapeutically relevant in this population (ie decrease fear conditioning and enhancing extinction).

Figure 5 has fear extinction twice in the schematic.

A trending interaction is used for post-hoc tests. Please describe in the methods what is significant and what is trending.

The methods have a section about pharmacological experiments, but those are not present in the manuscript.

Reviewer #2 (Remarks to the Author):

The authors present a compelling and very interesting set of data on the molecular response of CRH expressing neurons in the lateral aspect of the central amygdala (CeL) to fear expression and fear extinction. By using TRAP RNA sequencing, they identified a network of regulated genes especially following fear extinction that was associated with the upstream transcription factor CREB. Finally, the authors demonstrate that manipulation of CREB within the CeL CRH neurons leads to higher fear behavior during fear recall and extinction training. Overall, this study is well-conducted and the results are important. It is also noteworthy that the study was conducted in male and female mice. I have a number of questions that I would like the authors to clarify.

a. I am confused about the analysis of the RNAseq data. Did the authors use different cut-off criteria for the different comparisons? It is not clear to me why in the extinction vs tone (males and females) comparison there are so many differentially regulated genes following FDR correction, while there seem to be none in the fear conditioning vs tone comparison, although the reported p-values and fold changes are comparable. This needs to be clarified and described in greater detail.

b. I could not follow the reasoning of the authors regarding the weaker separation between the fear conditioning vs tone group. They argue that in the tone group there may be already sufficient stress due to transport / handling of the animals and the tone itself. However, the same would hold true for the comparison of extinction vs tone. As all the animals were sacrificed 2 hours after

the last manipulation, I don't see how this reasoning would explain why so many more differentially regulated genes were found in the fear extinction vs tone comparison (but see also my previous comment).

c. I missed a validation of the effect of altered CREB expression and – even more importantly – activity following fear expression and extinction. It would be crucial to show this, also in the context of the final experiment where the authors over-expressed CREB in the CeL.

d. Generally some of the figures use a very small font size (e.g. figure 3c, figure 4), which make them very hard to read. Please adjust.

e. Along the same lines, this reviewer is (as many other individuals) partially color blind, so labelling gene network modules by obscure colors is not really helpful. I am sure there are better ways to present the data and label the figures.

Reviewer #3 (Remarks to the Author):

McCullough et al use an elegant experimental design to probe the gene-molecular basis of fear conditioning and fear extinction in mice. They probe individual CRH expressing cells using translating ribosome affinity purification (TRAP) followed by RNA sequencing. They report that differential gene expression analyses performed following fear extinction learning demonstrate a fingerprint of broad decreases in neuronal activity of the CRH-expressing cells. Upstream gene set enrichment analyses suggested that the analysis profile associated with fear extinction was driven by reduction in the activity of the transcriptional regulator CREB. Accordingly, artificially increasing CREB expression promoted fear expression and suppressed fear extinction. The authors conclude that their data provide strong evidence for a pivotal role of CREB-mediated neuronal activation in the expression and extinction of fear.

The paper addresses an important broad conundrum in both basic science and clinical arenas of anxiety, depression and PTSD. The message is clear and the paper is well written. However, several issues deserve the authors' attention:

1. The mouse line used. There are several crh targeted mouse lines, and the congruence of the cre with native CRH expression in amygdala nuclei and nuclear subdivisions is variable in these lines. The authors use mRNA (RNAscope) to show that CRH cells exist in the CeL division of the central nucleus. However, they do not address CRH expression in the TRAPPED cells. E.g., an analysis filter ascertaining robust expression of CRH in the analyzed cells was not found by this reviewer. This topic should be discussed, and an estimate of the degree of congruence of endogenous peptide and the TRAPPED cells should be provided. Of interest, Figure 2D shows a huge variance in the expression of CRH in response to tone and fear conditioning (FC), consistent with a diverse cell population. (A supplementary table that was difficult to decipher identifies CRH as gene #~3700 in FC mice). In short, it is critical to recognize that some cells in the mice might not produce appreciable CRH, discuss this possibility and consider modest changes of the analyses that will increase confidence that bona fide CRH cells are being analyzed.

2. Minor additional notes regarding CRH in the analyses: It is slightly surprising that this gene does not come up more prominently in the modules. One notes that it is upregulated in the central amygdala by glucocorticoids rather than downregulated. In view of the fact that the gene is pivotal to the identity and presumed function of the CRH+ cells, and that there is a well described CREB dependent regulation of this gene, a deeper dig into this gene might be helpful.

3. The uncovering of CREB as an upstream regulator is slightly baffling. cAMP and CREB are intrinsic to the large majority of neuronal activity and might also be a marker of it rather than a driver?

3a. Were other potential TFs identified in the GSEA? For example, REST/NRSF, which also regulates crh expression under dynamic, physiological circumstances?

3b. CREB is ubiquitously expressed and regulates thousands of genes via divergent mechanisms. Thus, it may not be an intuitive candidate for the role assigned to it by the authors. Whereas viral-mediated INCREASE of CREB is supportive of the hypothesis, it is only partial proof. Did CREB phosphorylation (pCREB) increase in the transfected mice? Was CREB more located in the nucleus? Any changes to CBP? Additional easy studies such as IHC for pCREB on existing tissue might increase confidence in the authors' conclusions.

In summary, this is an excellent study addressing a topic of broad impact. The study will significantly benefit from a few enhancements as described above.

We would like to thank the editors and reviewers of our manuscript, **NCOMMS-20-04822-T**, “Genome-wide translational profiling of amygdala Crh-expressing neurons reveals role for CREB in fear extinction learning,” for their time and effort in reviewing this manuscript. We found the feedback extremely helpful.

Furthermore, we appreciate the many positive comments about the work, including, “Overall this is a very good paper”, “compelling and very interesting work”, “study is well-conducted and the results are important,” “elegant experimental design to probe the gene-molecular basis,” “data provide strong evidence for a pivotal role of CREB-mediated neuronal activation in the expression and extinction of fear,” and finally, “The paper addresses an important broad conundrum in both basic science and clinical arenas of anxiety, depression and PTSD.”

We also appreciate the constructive suggestions and comments, and we have made every effort to respond to the remaining concerns. To address the reviewer’s concerns, the revision includes new analyses, additional figures, corrected and updated figures, corrected text, and clarifications to the text along with expansion of discussion. Below, *in italics*, we provide the full reviewer’s comments and **in bold**, we provide a **point-by-point response** to each remaining reviewer concern below:

Reviewer #1:

Reviewer #1 (Remarks to the Author):

Crh-expressing neurons are found with the CeL of the CeA and have been implicated in fear expression and extinction. This paper uses TRAP-seq to investigate how translation changes in Crh neurons in the CeL following fear conditioning and extinction. DEGs following extinction, compared to tone exposure only, suggest a decrease in neuronal activity or plasticity and altered glucocorticoid receptor signaling. Interestingly, fear conditioning compared to tone only did not produce many DEGs, which appeared to be due to stress-related translational changes associated with behavioral testing itself. Upstream regulator analysis indicated CREB as a potential regulator of fear expression and extinction. This finding was validated with viral overexpression of CREB, which was sufficient to increase fear expression. Overall this is a very good paper with a few major issues. It is important to verify a few changes before its accepted: more support for explaining away sex differences in DEGs after extinction and a re-interpretation of their extinction behavior in their key validation experiment since it appears that the authors may be over-stating their results significantly.

Thank you for these helpful comments. Please find our point-by point responses below .

Major concerns:

Since males have significant DEGs, but females do not, the authors should consider an additional approach to help visualize the DEGs in females versus males after EXT. The correlations in the supplement are helpful to show that the sexes appear to be changing in a similar way, but an RRHO analysis or even a venn diagram comparing EXT v. TA in males and females would be an easier, more digestible, way to show that genes are being similarly up or down regulated in males and females. Alternatively, the authors could use a slightly less stringent cut-off and describe these genes as not surviving FDR correction, but very close to reaching significance. This is a critical point since DEGs are not significant in females, that could suggest

these genes are less important functionally or more variable. This brings up an important second point, these genes might be more variable in females due to cycling sex hormones. Did the authors verify estrous cycle phase in these female mice? Perhaps they were sacrificed at different phases leading to increased variability, making it more difficult to survive FDR correction.

Thank you for this important suggestion. In the revised submission, we have performed additional analyses and added additional figures to directly compare males and females and to attempt to control for estrous cycle by controlling for levels of an mRNA that is known to vary with estrous cycle stage.

As the reviewer suggested, we have now added in Fig. S2 – shown below – a Rank Rank Hypergeometric Overlap (RRHO) analysis comparing males and females, in addition to Spearman correlations of fold changes.

In an attempt to control for estrous cycle, by covarying for estrous state, we controlled all analyses for *Pgr* levels, a gene known to vary with estrus cycle. The correlations are presented here (Fig R1, below) and are essentially identical with the results presented in revised Fig. S2.

Figure R1. Spearman Correlation (down triangle) of fold-changes (log₂FC) and rank-rank hypergeometric overlap (upper triangle) of direction-signed p-values from the differential expression analyses (using *Pgr*)

levels as an additional covariate) based on between the three groups: Tone Alone/TA, Conditioning/FC and Extinction/EXT in both sexes: males (M) and females (F) and in across sexes (M&F).

In this initial study to examine cell-type-specific gene expression in Crh amygdala neurons, we included female mice without regard to estrous cycle phase. We agree this is an important point that will be addressed in future studies focusing on estrous-cycle-stage-specific differences, as well as potentially cell-type specific analyses of expressing neurons within amygdala. We have acknowledged this limitation in the revised discussion section.

In the site-specific viral experiments, sex is not discussed. Were both sexes used in this experiment? Since upstream analysis appears to have been conducted across sex it could be very important to confirm overexpression of CREB increases fear expression in both males and females independently.

All experimental mice for the CREB experiments were male. We agree that differential sex-specific experimental validation would be important in future studies, and we have acknowledged this limitation in the revised discussion section.

Furthermore, the suggestion that CREB overexpression blunts extinction learning is too strong. Subthreshold fear conditioning is used here in order to show increased fear expression in CREB overexpressing mice, but that results in a GFP group that doesn't show any fear extinction. CREB mice are showing fear expression (higher CS1 compared to preCS) and subsequent extinction (higher on CS1 and lower at CS15), but the GFP mice are not showing either. Therefore, extinction in CREB overexpressing mice artificially appears to be blunted. This "blunted" extinction is just a carry-over from these CREB mice showing fear expression in the first place when GFP controls do not and therefore have nothing to extinguish. This interpretation should be re-stated or an experiment should be added to confirm that extinction is in fact blunted under normal fear conditioning parameters, when control mice would should fear expression and extinction.

We agree that it is difficult to fully differentiate increased fear expression from blunted fear extinction with these experimental data. We have now clarified our interpretation in the revised manuscript, and further pointed out this limitation in the discussion section.

Minor concerns:

The authors should be sure to carefully check for typos and punctuation issues. There are several places where words appear to be missing.

Thank you, we have carefully re-read and edited where appropriate to correct remaining text errors.

In figure 1, the authors should consider removing panel b since it doesn't add very much to the figure and there is no non-DAPI comparison image included for the transgene.

The timeline (g) is also very confusing and makes it seem as though the EXT group were only exposed to tones. The authors should re-work this flowchart to make it clear that the EXT group was fear conditioned. For example, right justify or and tone alone and extend the arrow so it goes from FC to EXT.

Thank you, we believe the DAPI panel allows the reader to independently verify the location of expression within the CeA, we have updated Figure 1, improving timeline g, as suggested.

This caption also says gender instead of sex, which should be used in this case and is used in the rest of the manuscript.

Yes, thank you, we have made corrections to ensure that ‘sex’ as a biological variable in our mouse experiments, is used throughout and not ‘gender.’

In figure 3B it appears dark green is in the wrong place. Based on the text, it appears dark green should be a network that is overlapping between FC v TA and EXT v TA. That cell is also empty.

Thank you, we have corrected this figure accordingly.

Captions for Suppl. Fig 3 and 4 are the same.

Thank you for catching this, we have now corrected the captions for Supplemental Figures 3-4.

To what extent is CREB overexpressed?

We have addressed this in the discussion: “ The CREB construct used in over cell-type specific expression experiments has been used in a wide variety of experiments spanning decades by Carlezon and other labs. The Carlezon group has demonstrated that overexpression of CREB leads to increases in CREB mediated gene transcription, and CREB-mediated changes in electrophysiological responses. Additionally, they have demonstrated that CREB overexpression mimics pCREB-like activation of gene expression^{1, 2, 3}.”

In the site-specific viral experiment:

Sample sizes are not given for these behavioral experiments nor are ns statistics. If stats are given for ns fear conditioning, they should be given for EPC etc. Especially given the small differences that can be seen in these anxiety-like behaviors, like a decrease in center time in open field in CREB mice.

We appreciate these points and have added the sample sizes and non-significant statistic values where appropriate. Note that most of these sample sizes and statistics are located in the Figure Legend for Fig 5.

The authors should also discuss why they did not expect to find freezing in CREB overexpressing mice during this subthreshold fear conditioning protocol. If CREB mediates fear expression, why wouldn't freezing be increased during conditioning, as well as in the fear expression test.

We have now expanded our discussion of these results to address this question as follows, “Given that Pavlovian fear conditioning is explicitly an associative learning assay, over expression of CREB in Crh neurons would not be predicted to non-specifically increase freezing at baseline in the absence of a learned association; this prediction was supported by our results.”

The authors should also discuss whether CREB knockdown might be therapeutically relevant in this population (ie decrease fear conditioning and enhancing extinction).

Thank you, we have added text related to this interesting possibility in the discussion, as follows: “Cell-type specific targeting of CREB or other fear extinction related genes for knock-down, inhibition, or activation may reveal translationally relevant pathways for intervening in fear-related pathologies.”

Figure 5 has fear extinction twice in the schematic.

Thank you, this has now been corrected.

A trending interaction is used for post-hoc tests. Please describe in the methods what is significant and what is trending.

We have now updated the figure legend and methods to clarify the statistical definitions used for significance vs. trending within the results.

The methods have a section about pharmacological experiments, but those are not present in the manuscript.

We apologize for this oversight, and this section of the methods has now been removed.

Reviewer #2:

Reviewer #2 (Remarks to the Author):

The authors present a compelling and very interesting set of data on the molecular response of CRH expressing neurons in the lateral aspect of the central amygdala (CeL) to fear expression and fear extinction. By using TRAP RNA sequencing, they identified a network of regulated genes especially following fear extinction that was associated with the upstream transcription factor CREB. Finally, the authors demonstrate that manipulation of CREB within the CeL CRH neurons leads to higher fear behavior during fear recall and extinction training. Overall, this study is well-conducted and the results are important. It is also noteworthy that the study was conducted in male and female mice. I have a number of questions that I would like the authors to clarify.

Thank you for these helpful comments. Please find our point-by point responses below .

a. I am confused about the analysis of the RNAseq data. Did the authors use different cut-off criteria for the different comparisons? It is not clear to me why in the extinction vs tone (males and females) comparison there are so many differentially regulated genes following FDR correction, while there seem to be none in the fear conditioning vs tone comparison, although the reported p-values and fold changes are comparable. This needs to be clarified and described in greater detail.

Thank you for this point. We have further clarified the cut-off criteria used across the analyses in methods and results. It is important to note that the FDR Benjamini–Hochberg multiple testing correction of p-values will be different in different analysis based on different p-value distributions. FDR is the most common correction in gene expression studies. We have included this explanation in the discussion.

We were also surprised by the relative difference in identified DEGs in the extinction vs. tone condition compared to the Fear conditioning vs. tone condition. As further outlined below and in the text, we do not think this finding is a result of technical (e.g. RNA processing or DEG analysis) reasons. We do expect if we were to examine DEGs within Crh cells in fear vs. home cage and novel tone vs. home cage we might see similar effects, however such an additional experiment is out of scope of the current studies. Rather, we argue that Crh cells may be particularly sensitive to low levels of threat – thus not different between novel tone/context and footshock, but perhaps robustly different in the extinction – ‘threat expectancy’ condition and tone/context. Clearly more work needs to be done as is outlined in the discussion and limitations, but we believe the findings outlined here remain robust and important, at least for further understanding the nature of Crh cell-specific molecular regulation in fear extinction processes.

b. I could not follow the reasoning of the authors regarding the weaker separation between the fear conditioning vs tone group. They argue that in the tone group there may be already sufficient stress due to transport / handling of the animals and the tone itself. However, the same would hold true for the comparison of extinction vs tone. As all the animals were sacrificed 2 hours after the last manipulation, I don't see how this reasoning would explain why so many more differentially regulated genes were found in the fear extinction vs tone comparison (but see also my previous comment).

As discussed above, our interpretation relies on literature that suggests that Crh neurons participate in signaling low levels of threat and decrease their activity in response to fear extinction – see papers cited by Zweifel and Palmiter labs (in particular Sanford et al.,2017¹²). Thus, both transport and fear conditioning would lie at different magnitudes along the same ‘fear on’ behavioral and molecular expression vector while fear extinction would lie in the orthogonal ‘fear extinction or deactivation’ vector. Therefore, we would expect low and moderate threat states to have some similarities in their precipitated molecular expression profiles while we expect suppression behavioral states to have more divergent molecular expression profiles.

c. I missed a validation of the effect of altered CREB expression and – even more importantly – activity following fear expression and extinction. It would be crucial to show this, also in the context of the final experiment where the authors over-expressed CREB in the CeL.

Thank you for this point. As per our prior work with CREB overexpression discussed above we have addressed both expression and neuronal activity in text:

“The CREB construct used in over cell-type specific expression experiments has been used in a wide variety of experiments spanning decades by Carlezon and other labs. The Carlezon group has demonstrated that overexpression of CREB leads to increases in CREB mediated gene transcription, and CREB-mediated changes in electrophysiological responses. Additionally, they have demonstrated that

CREB overexpression mimics pCREB-like activation of gene expression^{1,2,3}.”

d. Generally some of the figures use a very small font size (e.g. figure 3c, figure 4), which make them very hard to read. Please adjust.

We appreciate this suggestion and have now updated the figures and fonts accordingly

e. Along the same lines, this reviewer is (as many other individuals) partially color blind, so labelling gene network modules by obscure colors is not really helpful. I am sure there are better ways to present the data and label the figures.

We appreciate this point – unfortunately, it is the standard nomenclature in the field examining gene coexpression networks. We have added a note to the figure legend to indicate that color names in Figure 3a are assigned by the WGCNA software, which is not under our control. We have worked to make the colors we use in Figure 3b/c are color blind compatible.

Reviewer #3:

Reviewer #3 (Remarks to the Author):

McCullough et al use an elegant experimental design to probe the gene-molecular basis of fear conditioning and fear extinction in mice. They probe individual CRH expressing cells using translating ribosome affinity purification (TRAP) followed by RNA sequencing. They report that differential gene expression analyses performed following fear extinction learning demonstrate a fingerprint of broad decreases in neuronal activity of the CRH-expressing cells. Upstream gene set enrichment analyses suggested that the analysis profile associated with fear extincti was driven by reduction in the activity of the transcriptional regulator CREB. Accordingly, artificially increasing CREB expression promoted fear expression and suppressed fear extinction. The author s conclude that their data provide strong evidence for a pivotal role of CREB-mediated neuronal activation in the expression and extinction of fear.

The paper addresses an important broad conundrum in both basic science and clinical arenas of anxiety, depression and PTSD. The message is clear and the paper is well written. However, several issues deserve the authors' attention:

Thank you for these helpful comments. Please find our point-by point responses below .

1. The mouse line used. There are several crh targeted mouse lines, and the congruence of the cre with native CRH expression in amygdala nuclei and nuclear subdivisions is variable in these lines. The authors use mRNA (RNAscope) to show that CRH cells exist n the CeL division of the central nucleus. However, they do not address CRH expression in the TRAPPED cells. E.g., an analysis filter ascertaining robust expression of CRH in the analyzed cells was not found by this reviewer. This topic should be discussed, and an estimate of the degree of congruence of endogenous peptide and the TRAPPED cells should be provided. Of interest, Figure 2D shows a huge variance in the expression of CRH in response to tone and fear conditioning (FC), consistent with a diverse cell population. (A supplementary table that was difficult to decipher identifies CRH

as gene #~3700 in FC mice). In short, it is critical to recognize that some cells in the mice might not produce appreciable CRH, discuss this possibility and consider modest changes of the analyses that will increase confidence that bona fide CRH cells are being analyzed.

Thank you for these important points. We have taken several steps to address these concerns:

- 1) **Methods section:** "Both Crh-cre lines have previously been validated to have accurate targeting of Cre expression to Crh neurons in the CeA^{4,5,6}."
- 2) Thank you for this point. As stated, we appreciate and recognize that some cells in the mice might not produce appreciable CRH protein or *Crh* mRNA, and we have further discussed this possibility in discussion and limitations. CRH and *Crh* expression itself is known to be dynamic in amygdala as a function of behavioral state, as noted from prior work from our group and others^{7,8,9,10}, such that level of expression of CRH or *Crh* within the Crh-TRAP cells is not likely to be a static marker of cell identity¹⁰. We have now expanded our discussion of CRH regulation within the *Discussion* section as follows: "It is important to note that the *Crh* expressing population is dynamic and while transgenic mouse line used reliably captures the Crh neuron population, *Crh* expression may vary between cells depending on behavioral state¹⁰."
- 3) Figure 2D, as stated in the manuscript, is a qPCR from whole tissue punch. These results indicate variability in tissue wide *Crh* expression and have little bearing on the diversity of the cell population.
- 4) We have thus provided further discussion of the level of CRH expression overlap in the Crh-TRAP cells to the extent to which we are able.
- 5) We want to point out that in Crh neurons, *Crh* is not supposed to be the gene with highest average expression. It is rather a marker gene, but not the top expressed gene. We used data from one of our recent amygdala bulk RNA-seq papers¹¹ and calculated the ratio of *Crh* average expression to the highest expressed gene (i.e., *Hspa5*). This ratio was 0.027, while in our present TRAP-seq study this ratio is 0.69 (26 times higher) indicating a significant enrichment for *Crh* expression.

2. *Minor additional notes regarding CRH in the analyses: It is slightly surprising that this gene does not come up more prominently in the modules. One notes that it is upregulated in the central amygdala by glucocorticoids rather than downregulated. In view of the fact that the gene is pivotal to the identity and presumed function of the CRH+ cells, and that there is a well described CREB dependent regulation of this gene, a deeper dig into this gene might be helpful.*

Thank you for this point. We have addressed the identity CRH neurons in the comment above. We apologize that it is somewhat unclear to us what the reviewer is suggesting with regards to 'a deeper dig into this gene might be helpful', as there is a large literature on the role of Crh more broadly on stress and fear responses, in addition to multiple studies from our group (e.g. ¹³⁻²⁰). Additionally, as discussed we agree that there is likely a direct link between CREB and *Crh* expression. *Crh* does appear in at least one of the identified WGCNA modules. Future experiments will continue to explore the role of CREB and CRH in fear and fear extinction learning .

3. *The uncovering of CREB as an upstream regulator is slightly baffling. cAMP and CREB are intrinsic to the large majority of neuronal activity and might also be a marker of it rather than a driver?*

3a. Were other potential TFs identified in the GSEA? For example, REST/NRSF, which also regulates crh expression under dynamic, physiological circumstances?

Thank you for this point. Our interpretation of CREB activation, was less that it was specific to these cells only, but more that it was representative of robust molecular markers of plasticity during extinction processing in the Crh cell population. While other TFs were also identified in the GSEA, as suggested, none were as robust as CREB in identified pathways. We have added further discussion of these possibilities in the Discussion section, accordingly.

3b. CREB is ubiquitously expressed and regulates thousands of genes via divergent mechanisms. Thus, it may not be an intuitive candidate for the role assigned to it by the authors. Whereas viral-mediated INCREASE of CREB is supportive of the hypothesis, it is only partial proof. Did CREB phosphorylation (pCREB) increase in the transfected mice? Was CREB more located in the nucleus? Any changes to CBP? Additional easy studies such as IHC for pCREB on existing tissue might increase confidence in the authors' conclusions.

Each of these considerations is very important for understanding the activity and trafficking of CREB *in vivo*. We and many others have extensively validated this expression vector and described the expression, molecular, and activity related changes resulting from CREB over expression.

We have attempted to address these concerns in the discussion as follows: “ The CREB construct used in over cell-type specific expression experiments has been used in a wide variety of experiments spanning decades by Carlezon and other labs. The Carlezon group has demonstrated that overexpression of CREB leads to increases in CREB mediated gene transcription, and CREB-mediated changes in electrophysiological responses. Additionally, they have demonstrated that CREB overexpression mimics pCREB-like activation of gene expression ^{1, 2, 3}.”

Additional characterization of the role that CREB plays in CRH neurons during fear and fear extinction learning will be an important part of future studies.

In summary, this is an excellent study addressing a topic of broad impact. The study will significantly benefit from a few enhancements as described above.

Thank you again for your supportive comments.

References for Response to Reviewers:

1. Carlezon WA, *et al.* Regulation of cocaine reward by CREB. *Science* 282, 2272-2275 (1998).
2. Sakai N, *et al.* Inducible and brain region-specific CREB transgenic mice. *Molecular pharmacology* 61, 1453-1464 (2002).
3. Dong Y, *et al.* CREB modulates excitability of nucleus accumbens neurons. *Nat Neurosci* 9, 475-477 (2006).
4. Gafford G, Jasnow AM, Ressler KJ. Grin1 receptor deletion within CRF neurons enhances fear memory. *PLoS One* 9, e111009 (2014).
5. Martin EI, *et al.* A novel transgenic mouse for gene-targeting within cells that express corticotropin-releasing factor. *Biol Psychiatry* 67, 1212-1216 (2010).
6. Chen Y, Molet J, Gunn BG, Ressler K, Baram TZ. Diversity of Reporter Expression Patterns in Transgenic Mouse Lines Targeting Corticotropin-Releasing Hormone-Expressing Neurons. *Endocrinology* 156, 4769-4780 (2015).
7. Marini F, *et al.* Single exposure to social defeat increases corticotropin-releasing factor and glucocorticoid receptor mRNA expression in rat hippocampus. *Brain Res* 1067, 25-35 (2006).
8. Tran L, Greenwood-Van Meerveld B. Altered expression of glucocorticoid receptor and corticotropin-releasing factor in the central amygdala in response to elevated corticosterone. *Behav Brain Res* 234, 380-385 (2012).
9. Merali Z, McIntosh J, Kent P, Michaud D, Anisman H. Aversive and Appetitive Events Evoke the Release of Corticotropin-Releasing Hormone and Bombesin-Like Peptides at the Central Nucleus of the Amygdala. *J Neurosci* 18, 4758-4766 (1998).
10. Shepard JD, Barron KW, Myers DA. Corticosterone delivery to the amygdala increases corticotropin-releasing factor mRNA in the central amygdaloid nucleus and anxiety-like behavior. *Brain Res* 861, 288-295 (2000).
11. Lori A, Maddox SA, Sharma S, Andero R, Ressler KJ, Smith AK. Dynamic Patterns of Threat-Associated Gene Expression in the Amygdala and Blood. *Front Psychiatry* 9, 778 (2018).
12. Sanford CA, *et al.* A Central Amygdala CRF Circuit Facilitates Learning about Weak Threats. *Neuron* 93, 164-178 (2017).
13. Lebow MA, Schroeder M, Tsoory M, Holzman-Karniel D, Mehta D, Ben-Dor S, Gil S, Bradley B, Smith AK, Jovanovic T, Ressler KJ, Binder EB, Chen A. Glucocorticoid-induced leucine zipper "quantifies" stressors and increases male susceptibility to PTSD. *Transl Psychiatry*. 2019 Jul 25;9(1):178.

14. Dedic N, Kühne C, Jakovcevski M, Hartmann J, Genewsky AJ, Gomes KS, Anderzhanova E, Pöhlmann ML, Chang S, Kolarz A, Vogl AM, Dine J, Metzger MW, Schmid B, Almada RC, Ressler KJ, Wotjak CT, Grinevich V, Chen A, Schmidt MV, Wurst W, Refojo D, Deussing JM. Chronic CRH depletion from GABAergic, long-range projection neurons in the extended amygdala reduces dopamine release and increase anxiety. *Nat Neurosci*. 2018 Jun;21(6):803-807.
15. Chen Y, Molet J, Gunn BG, Ressler K, Baram TZ. Diversity of Reporter Expression Patterns in Transgenic Mouse Lines Targeting Corticotropin-Releasing Hormone-Expressing Neurons. *Endocrinology*. 2015 Dec;156(12):4769-80.
16. Hurt RC, Garrett JC, Keifer OP Jr, Linares A, Couling L, Speth RC, Ressler KJ, Marvar PJ. Angiotensin type 1a receptors on corticotropin-releasing factor neurons contribute to the expression of conditioned fear. *Genes Brain Behav*. 2015 Sep;14(7):526-33.
17. Gafford G, Jasnow AM, Ressler KJ. Grin1 receptor deletion within CRF neurons enhances fear memory. *PLoS One*. 2014 Oct 23;9(10):e111009.
18. Gafford GM, Guo JD, Flandreau EI, Hazra R, Rainnie DG, Ressler KJ. Cell-type specific deletion of GABA(A) α 1 in corticotropin-releasing factor-containing neurons enhances anxiety and disrupts fear extinction. *Proc Natl Acad Sci U S A*. 2012 Oct 2;109(40):16330-5.
19. Flandreau EI, Ressler KJ, Owens MJ, Nemeroff CB. Chronic overexpression of corticotropin-releasing factor from the central amygdala produces HPA axis hyperactivity and behavioral anxiety associated with gene-expression changes in the hippocampus and paraventricular nucleus of the hypothalamus. *Psychoneuroendocrinology*. 2012 Jan;37(1):27-38.
20. Keen-Rhinehart E, Michopoulos V, Toufexis DJ, Martin EI, Nair H, Ressler KJ, Davis M, Owens MJ, Nemeroff CB, Wilson ME. Continuous expression of corticotropin-releasing factor in the central nucleus of the amygdala emulates the dysregulation of the stress and reproductive axes. *Mol Psychiatry*. 2009 Jan;14(1):37-50.

REVIEWERS' COMMENTS:

Reviewer #1 (Remarks to the Author):

The authors have addressed the prior concerns.

Reviewer #2 (Remarks to the Author):

I thank the authors for clarifying and addressing my previous criticism. There are no remaining issues from my side.

Reviewer #3 (Remarks to the Author):

In this thoughtfully revised version of the manuscript, the authors have largely addressed my comments and suggestions.

TZB